# Climate Stressors and Physiological Dysregulations: Mechanistic Connections to Pathologies

**DOI:** 10.3390/ijerph21010028

**Published:** 2023-12-23

**Authors:** Hajar Heidari, David A. Lawrence

**Affiliations:** 1Department of Biomedical Sciences, University at Albany School of Public Health, Rensselaer, NY 12144, USA; hheidari@albany.edu; 2Department of Environmental Health Sciences, University at Albany School of Public Health, Rensselaer, NY 12144, USA; 3Wadsworth Center, New York State Department of Health, Albany, NY 12208, USA

**Keywords:** folate, oxidative stress, autoimmune disease, mitochondria, metabolites, chaperones

## Abstract

This review delves into the complex relationship between environmental factors, their mechanistic cellular and molecular effects, and their significant impact on human health. Climate change is fueled by industrialization and the emission of greenhouse gases and leads to a range of effects, such as the redistribution of disease vectors, higher risks of disease transmission, and shifts in disease patterns. Rising temperatures pose risks to both food supplies and respiratory health. The hypothesis addressed is that environmental stressors including a spectrum of chemical and pathogen exposures as well as physical and psychological influences collectively impact genetics, metabolism, and cellular functions affecting physical and mental health. The objective is to report the mechanistic associations linking environment and health. As environmental stressors intensify, a surge in health conditions, spanning from allergies to neurodegenerative diseases, becomes evident; however, linkage to genetic-altered proteomics is more hidden. Investigations positing that environmental stressors cause mitochondrial dysfunction, metabolic syndrome, and oxidative stress, which affect missense variants and neuro- and immuno-disorders, are reported. These disruptions to homeostasis with dyslipidemia and misfolded and aggregated proteins increase susceptibility to cancers, infections, and autoimmune diseases. Proposed interventions, such as vitamin B supplements and antioxidants, target oxidative stress and may aid mitochondrial respiration and immune balance. The mechanistic interconnections of environmental stressors and disruptions in health need to be unraveled to develop strategies to protect public health.

## 1. Introduction

The significant influence of environmental stressors on humans is undeniable. The environment in our ever-evolving world has a profound impact on shaping well-being throughout the entire lifespan [1]. This review presents the multifaceted interplay of environmental factors, toxicological effects, and the bidirectional influences of exogenous and endogenous factors on human health (Figure 1). As there is no one environmental stressor affecting health, there is also no one organ system involved. Numerous types of environmental stressors disrupt bidirectional communications among organ systems, which are needed to maintain homeostasis. The diversity of environmental stressors occurring with climate change affects physical and mental health [2]. There have been many reports on climate changing the environment and exposure to different environmental stressors provoking physical and mental disorders; the associations of physical and psychological regulations have merging mechanisms [3,4,5,6]. Although there has been an exponential increase in reports linking environment and health, few reports link health outcomes to exogenous initiators and endogenous mechanisms responsible for the paths toward ill health. Herein, the focus is on environmental stress, especially heat stress, on human health due to posited cellular and molecular consequences of oxidative stress (OS) and altered immunity. 

Climate extremes of heat, drought, and rain trigger a complex cascade of effects, disrupting the behavior of disease vectors and posing formidable challenges to public health [7,8]. Rising temperatures drive changes in plant growth patterns and heighten pollen production, leading to issues in food supply and quality and respiratory health [9]. Some foods may have increased fungi levels producing mycotoxins such as 3-nitroproprionic acid (3-NPA), which causes mitochondrial dysfunction and neuropathologies [10]. More frequent natural disasters like hurricanes and wildfires can also result in injuries and physical and psychological trauma, all of which contribute to the escalation of health concerns [3]. Prolonged heatwaves strain the cardiovascular system and damage epithelial, endothelial, and immune cells [11,12]. Extreme weather events also affect mental health, which may relate to anxiety mediating neuroimmune effects and amplifying health concerns [13].

Our environment is a demonstrable web of stressors casting a net to encase health with the evolving exposome [14]. The stressors include chemicals (organic and inorganic toxins and toxicants) such as benzene and lead, pathogens (viruses, bacteria, fungi, and worms) and the new global spread of transmitting vectors, e.g., mosquitoes [8,15], physical modifiers, e.g., heat and radiation, and psychological and socioeconomic effects heightening anxiety and depression [16,17,18]. Differential lifetime exposures to these stressors craft health disparities [19]. Stress in children from life-disrupting events can have profound long-term influences. Increased stress levels due to prolonged exposure to environmental stressors intricately mold the landscape of our physical and mental health [20]. As environmental stressors intensify in the face of climate change, we witness a marked surge in health conditions, spanning from asthma and allergies to cardiovascular diseases, neurodevelopmental disorders like autism spectrum disorders (ASD), and neurodegenerative diseases such as Parkinson’s disease and Alzheimer’s disease [21,22]. Here, we introduce a hypothesis that ties climate-induced OS to the realm of neuro- and immuno-disorders emphasizing the critical role of OS in shaping health outcomes.

This review underscores the intricate connections between metabolic and mitochondrial dysfunction, OS, and the differential modulation of immune cells and their activities along with the involvement of OS-associated genetic variants [23] and possible protein variants due to genetic mutations in protein-coding regions [24]. The inflammation and immune imbalances induced by environmental stressors elevate our susceptibility to cancers, infections, and autoimmune diseases [25,26]. With the dichotomy of inducers and responses to these challenges, there are proposed interventions, including vitamin B supplements and antioxidants, to enhance glutathione (GSH) synthesis and alleviate OS, which is discussed as an underlying cause of cell damage and illnesses [27,28]. Improving the redox balance will aid in immune defenses against pathogens and lessen the development of altered and dysfunctional self-molecules and cells. With less presence of damaged self-molecules and cells, there would be a lower incidence of autoimmune diseases. The aim of an appropriate immune system is to maintain “immunohomeostasis”, a regulated balance of aggression, and the removal of foreignness (pathogens and cancers) and damaged and senescent cells and immunity with normal levels of functioning molecules and cells [29]. Environmental stressors disrupting immunohomeostasis are described, and metabolic and cellular interconnecting pathways, such as folate/methionine cycles, are proposed to be involved. The enhanced prevalence of diseases by environmental stressors and some endogenous levels of enhancers and inhibitors needed for health are defined and illustrated. For example, endogenous homocysteine (Hcy) is a normal metabolite of the methionine cycle, but hyperhomocysteinemia is associated with many diseases like coronary artery disease and stroke, neurological disorders such as Alzheimer’s disease and cognitive decline, birth defects like neural tube defects, bone health problems including osteoporosis and fractures, and certain cancers and psychiatric disorders [30]. Methylenetetrahydrofolate reductase (MTHFR) gene variations affect the levels of Hcy and GSH [31] and influence immunity [32], which highlights the intricate web of genetics, nutrition, metabolites, and dysfunctional outcomes that may occur with heat stress (Figure 2). Uncovering metabolic processes connected to aberrant immunity and ill health needs investigation to guide future public health interventions and mitigate stressors that disrupt health.

The literature searches using PubMed, Google, ResearchGate, and Scopus included various combinations of the following terms: oxidative, oxidant, climate, environmental, environment, helper T cells, immune, immunity, autism, autoimmune, inflammation, glutathione, thiols, methionine, folate, metabolic syndrome, B vitamins, and chaperones. Only reports that mechanistically connected two or more of the terms were included. Some of the suggested connections also came from research related to autoimmunity, autism, and responses to infections from acknowledged NIH support. 

## 2. Climate Change Impact on the Exogenous and Endogenous Environment

The connections between environmental factors and human health emphasize the complex influences of water, air, and soil quality as their constituents may have variant and profound cellular and molecular effects on sustainability and well-being [33,34]. Understanding the chemistry of our surroundings, the toxicological effects of environmental stressors, and their intricate relationship with human health is not only essential for our present health but also for coping with future changes. This imperative takes on a heightened significance when viewed in the broader context of climate change, a global phenomenon that exerts multifaceted influences on various facets of human health [35,36]. A new era of health-related challenges started with climate change, which is driven by factors such as industrialization, deforestation, and the release of greenhouse gases [37,38]. With these changes, we are not only witnessing shifts in disease vector behaviors and increased disease transmission risks but also a reshaping of disease landscapes. In parallel, rising temperatures have caused transitions in plant growth patterns, leading to food scarcity in specific locations as well as upsurged production of weeds and pollens, which has caused the use of more herbicides and pesticides. This exacerbates challenges related to food supply and triggers respiratory allergies and exposures to toxicants [39,40]. Furthermore, the increased frequency and severity of extreme weather events expose all animals to both physical stressors like heat and psychological stressors such as anxiety, gradually eroding physical and mental well-being [41]. With environmental exposures, there are many factors that stress our bodies, like chemicals, germs, and airborne particles. These factors have an impact on both our physical health and our mental well-being [36]. As climate change becomes worse, we see more health problems like asthma, allergies, cardiovascular diseases, metabolic syndrome, and different brain-related conditions, such as ASD, Parkinson’s disease, and Alzheimer’s disease [42,43,44,45]. While we are improving diagnosis of these health issues, there’s an important link between climate changes and associated physical and psychological effects on the microbiome-gut-brain axis and neuroendocrine immune network elevating OS, which affects all cells and tissues throughout the body [44,45] that needs consideration and research.

### 2.1. Heat and OS

As heatwaves become more frequent and severe, there is growing concern about increased heat-related health problems and deaths, particularly among children, the elderly, those with chronic illnesses, and underserved communities with limited access to resources and healthcare [46,47]. In the face of intensifying climate change, frequent exposure to heat stress necessitates greater blood circulation to extremities to facilitate cooling through sweating. However, this heightened demand for energy in the form of adenosine triphosphate (ATP) for cardiovascular work concomitantly reduces blood supply to certain organs making the heart pump faster [48,49]. The metabolic shift for more ATP elevates the concomitant increase in ROS, resulting in OS, which compromises cellular functions and can lead to cell death [50,51]. OS disrupts the intricate network of the gut-brain axis and neuroendocrine immune interactions, potentially leading to leaky gut syndrome and enhanced inflammation [52,53].

### 2.2. Commonality of OS with Pathologies and Altered Immunity

At the molecular level, ROS are central players in the intricate interplay between environmental stressors and health outcomes. One of the consequences arising from an imbalance between the production of ROS and the lack of antioxidant defenses, such as GSH, to neutralize ROS, is OS [54]. Under the influence of environmental stressors such as heat and exposure to toxins and toxicants, cells and tissues experience heightened ROS production. OS can lead to oxidative damage to cellular components, including lipids, proteins, and DNA. 

Immune cell imbalances and OS have been observed in various health conditions and in response to various environmental stressors such as metals [55] like lead [56], “cold restraint” (mouse in a 50 mL refrigerated (8 °C) tube with holes for 1 h) [57], dietary factors [58], exposure to air pollutants [59], radiation exposure [60], and heat [61]. The physical/psychological stress of mice with cold restraint caused sympathetic nerve release of norepinephrine, which inhibits immune defense [57], and norepinephrine induces ROS [62]. Interestingly, many of the environmental stressors affecting immune imbalances and neuropathologies are connected to the gut–brain axis and type 2 inflammation [63,64]. Many of the inducers of immune imbalance relate directly or indirectly to OS, e.g., with fewer protein and non-protein thiols, such as GSH, there is less helper T cell (Th) activation of the Th1 subset and more activation of Th2 [65]. Less Th1 activity affects defenses against infections and cancers. With enhanced Th2 activity and possibly more Th17 and less regulatory T cell (Treg) activity, there is more type 2 inflammation, which helps to fight parasites. Th1 activity leads to inflammation, and Th2 activity is often considered anti-inflammatory. However, when exaggerated type 2 inflammation occurs (Figure 3), it may lead to autoimmune diseases, asthma, and/or allergies with increased responses from eosinophils and antibody production. Increased eosinophil levels can lead to allergies and skin rashes. Increased antibody production may include autoantibodies and immune complexes leading to cell damage and possible autoimmune diseases. 

It is important to consider environmental stressors regarding the development of autoimmune diseases and type 2 inflammation with and without allergies, particularly in the context of the portions of activated Th1, Th2, Th17, and Treg cells. The Th1:Th2 immune response ratio affects the balance between cell-mediated and antibody-mediated and inflammatory and anti-inflammatory responses, respectively, as has been observed with human immunodeficiency virus (HIV) infection leading to acquired immune deficiency syndrome (AIDS) [65]; this imbalance is due to a loss of GSH [66]. The Th1:Th2 imbalance with the loss of GSH, which is associated with aging, HIV, and inflammation, is displayed in Figure 3. An imbalance between T cell subsets also includes regulatory T cells (Treg) and Th17 cells; a Treg and Th17 imbalance affects cardiovascular and intestinal disorders with increases in OS [67,68].

In the case of MIS-C, there is a notable dysregulation of immunity with inflammation and OS [69], which contributes to cardiovascular dysfunction. Additionally, an elevated Neutrophil-Lymphocyte Ratio that induces OS [70] has been observed in patients, reflecting an increased inflammatory state, which underscores the complexity of the immune changes affecting neurological symptoms [71]. Understanding these immune response imbalances is vital for both diagnosis and treatment strategies. An increased prevalence of allergies has been connected to immune responses influencing neurodevelopmental disorders like ASD [72], which suggests a possible link between immune dysregulation, allergies, and conditions like ASD. Associations with immune system imbalances, allergies, and type 2 inflammation have also been observed in conditions like SLE, asthma, and neurodegenerative disorders such as Parkinson’s disease and Alzheimer’s disease. Interestingly, Parkinson’s disease and Alzheimer’s disease have been suggested to have pathology related to autoantibodies [73,74,75]. Autoantibody to stress-inducible phosphoprotein 1 (STIP1), a co-chaperone of chaperone heat shock protein 90 (HSP90), increases Parkinson’s disease [75]. It has recently been reported that the modification of succinate dehydrogenase (Complex II) in the electron transport chain of mitochondria can lead to enhanced presentation of antigen to killer T cells [76], which might involve the missense mitochondrial gene or protein changes leading to autoimmune induction. The intricate network of connections among immune responses, allergies, and various health conditions highlights the necessity for investigations to research the involved fundamental mechanisms.

In the past decade, we have witnessed a rising prevalence of allergies, autoimmune disorders, and imbalances in immune responses, posing a significant challenge to public health. It is essential to investigate the intricate connections between these factors and their potential impact on a wide range of health conditions, including SARS, MIS-C, ASD, SLE, asthma, Parkinson’s disease, and Alzheimer’s disease. Shedding light on these relationships may offer valuable insights into the development of effective diagnostic and therapeutic approaches for these complex and interrelated health issues.

### 2.3. OS and Neurological Disorders

The molecular pathways of OS provide insights into general health outcomes and offer crucial perspectives on the development of neurodevelopmental disorders like ASD. OS-induced damage to cellular components, including DNA, can significantly impact neurodevelopment [77]. For instance, DNA damage in developing neural cells may lead to genetic mutations or epigenetic alterations, disrupting the precise orchestration of neurodevelopmental processes [78]. These disruptions can manifest as neurodevelopmental disorders, with ASD being a prime example. Recent research has uncovered compelling links between oxidative stress and the pathogenesis of ASD. Oxidative damage to critical biomolecules, such as proteins and nucleic acids, can influence synaptic plasticity and neural connectivity, which are vital processes in neurodevelopment [79]. Moreover, OS-induced inflammation has been implicated in the etiology of ASD [80]. Inflammatory responses triggered by OS can lead to the activation of microglia and astrocytes, contributing to neuroinflammation, which has been observed in individuals with ASD [81]. These molecular connections highlight how environmental stressors, through OS, contribute to the onset and progression of ASD. 

A recent report sheds light on the involvement of Wiskott–Aldrich Syndrome Protein Family Member 3 (WASF3) expression with the complex pathophysiology of myalgic encephalomyelitis/chronic fatigue syndrome (ME/CFS) [82], and its influence on endoplasmic reticulum (ER), OS, and mitochondrial dysfunction [83]. This emphasizes the organelle associations that connect ER stress with inflammation and OS [84], a notorious cellular phenomenon for its disruptive effects on normal cellular functions, and it further unveils a possible connection between genetic factors, ER stress, and the multifaceted symptoms characteristic of ME/CFS. Understanding these associations is of significance, given the challenging nature of ME/CFS and the need to decipher its underlying mechanisms for the development of effective therapeutic interventions. The study’s exploration of the relationship between the *WASF3* gene and ER stress could extend its relevance beyond the realm of ME/CFS. This genetic influence on organelles and their interactions might have implications for individuals grappling with Long-COVID, a condition marked by persistent symptoms after COVID-19 infection. The study’s focus on participants with a history of cancer, chronic fatigue, and autoimmune disorders underscores the intricate interplay between genetic predisposition and environmental influences in the context of various health conditions. These new findings pave the way for further research regarding exercise intolerance and fatigue-related disorders mechanisms, offering promising insights into the management and treatment of these complex health challenges.

One possible explanation for the higher incidence of Alzheimer’s disease and Parkinson’s disease among older individuals is that OS tends to increase with age. This, much like in the case of ASD, has been linked to changes in the microbiome-gut-brain axis [85,86]. Heat stress has been related to OS-inducing gut changes, and the detrimental influences can be reduced with antioxidants such as vitamin C, vitamin E, and GSH [87] and metabolites affecting the folate and methionine cycles [88]. 

### 2.4. OS and Cardiovascular Disorders

ROS-induced oxidative damage plays a pivotal role in the development and progression of cardiovascular diseases [89], which may connect to other diseases. The oxidative modification of lipids, especially low-density lipoproteins (LDL), contributes to the formation of atherosclerotic plaques, a hallmark of cardiovascular disorders like atherosclerosis [90]. Moreover, OS-induced inflammation can promote endothelial dysfunction, further exacerbating cardiovascular disease [89,91].

### 2.5. Genetic Variations and Susceptibility

OS induces DNA damage and can give rise to genetic mutations and instability, potentially contributing to the progression of various health conditions [92]. Furthermore, while OS is known to cause significant damage to cellular components, including lipids, proteins, and DNA, it is essential to recognize that not all cellular defense mechanisms are equally robust in the face of these stressors. Recent research has highlighted the vulnerability of certain DNA repair enzymes to OS [93]. These enzymes play a pivotal role in maintaining genomic integrity by correcting DNA damage resulting from various insults, including oxidative stress-induced lesions [94]. However, when exposed to high OS, some DNA repair enzymes may experience reduced efficiency, potentially compromising their ability to repair damaged DNA strands accurately [93]. This sensitivity raises intriguing questions about the interplay between oxidative stress, DNA repair mechanisms, and genomic stability, warranting further investigation to elucidate its precise implications for health and disease.

Genetics plays a pivotal role in shaping an individual’s vulnerability to environmental stressor-induced health consequences. Variations in specific genes can influence an individual’s ability to detoxify harmful substances, repair DNA damage, or mount an effective immune response. For example, the *MTHFR* gene, which encodes an enzyme involved in folate metabolism, has been implicated in various health conditions. Genetic variations in *MTHFR* can lead to elevated homocysteine levels, which have been associated with increased risk for cardiovascular diseases and neurodevelopmental disorders like ASD [95,96]. Climate-related heat stress could make individuals with *MTHFR* mutations more susceptible to the folate cycle’s need to interface with the methionine cycle, which is required for methylation processes (Figure 2). Understanding these genetic variations and their interactions with environmental stressors is essential for identifying vulnerable populations and tailoring preventive measures. Specific genetic polymorphisms of the detoxification enzymes glutathione S-transferases (GSTs) can significantly influence an individual’s ability to metabolize and eliminate environmental toxins. 

The emerging field of epigenetics reveals how environmental stressors can induce epigenetic modifications, including DNA methylation, histone alterations, and microRNA expression changes, which, in turn, alter gene expression patterns and contribute to the development of various health conditions [97]. Understanding gene–environment interactions is crucial, as individuals with certain genetic variants may have differential susceptibility to the health impacts of environmental stressors. In ASD or Intellectual Disability (ID), gene changes responsible for epigenetic mechanisms have been observed, e.g., a mutation of *HIST1H1E*, which encodes the H1 histone linker protein affecting gene transcription regulation, resulted in a reduction in protein expression [98]; for 215 genes associated with ASD, 42 genes (19.5%) were reported to be modified through epigenetic mechanisms [99]. In a study on heat-stressed Holstein calves, 8567 of their genes had modified expression, with 465 (5.4%) being increased and 49 (0.57%) being decreased in expression [100]. In an extreme heat situation, humans had rapid and significant changes in the expression patterns of genes and the suppression of more than two-thirds of differentially expressed genes [101]. Stress-related signaling pathways, immune response pathways, and metabolic processes were affected by these gene alterations, particularly immune function-related genes like *NFKB1*. With epigenetic changes, transgenerational effects of stresses can be passed on for generations, which highlights the far-reaching implications of genetic responses to environmental challenges [102]. Genetic biomarkers are also emerging as valuable tools for assessing susceptibility to environmental stressors, aiding in the identification of at-risk individuals or populations. Advancements in gene editing technologies and gene-based therapeutics offer promising avenues for mitigating the genetic effects of environmental stressors and tailoring preventive measures [103].

It is essential to understand that the mere presence of a ‘pathogenic’ allele in an individual does not guarantee the onset of clinical disease, given the multifaceted interplay of factors like genetic background and environmental influences. Within the extensive pool of over 4 million observed missense variants, approximately only 2% have been definitively categorized as pathogenic or benign, leaving the majority with an uncertain clinical significance [104]. To tackle this challenge, machine-learning tools, like AlphaMissense, aid in the integration of gene-level predictions with population-based methodologies and pathologies [104]. This synergistic approach quantitatively assesses the functional relevance of genes, especially in the context of shorter human genes where cohort-based techniques often encounter statistical limitations. More intricate molecular analyses of the consequences of variants on protein function may facilitate the identification of pathogenic missense mutations, unveil hitherto unknown disease-associated genes, and ultimately amplify the diagnostic potential for rare genetic disorders.

### 2.6. Folate and Methionine Cycles

Investigation into the intricate folate and methionine cycles unveils a revealing connection between autoimmunity, inflammation, and complex health conditions such as ASD [105] and MIS-C [106]. Homocysteine, a natural amino acid derived from dietary protein sources, assumes a central role in these biochemical processes [107]. While homocysteine serves as an intermediary in essential pathways, its regulation is of paramount importance. Elevated levels, known as hyperhomocysteinemia, have been linked to an increased risk of cardiovascular diseases, cognitive impairment, and neurodevelopmental disorders [108,109,110].

Hyperhomocysteinemia, a common thread in the contexts of both ASD and MIS-C, introduces a novel dimension to our understanding of these conditions [111]. In ASD, elevated homocysteine levels are linked to disruptions in gene regulation during critical brain development stages, potentially contributing to neurodevelopmental abnormalities [95,96]. Conversely, in MIS-C, the intriguing association between homocysteine and inflammation sheds new light on the underlying mechanisms of this immune-mediated condition. Investigating homocysteine levels emerges as a valuable diagnostic avenue for both disorders, offering insights into early detection and monitoring. Moreover, unraveling the intricate mechanisms through which homocysteine influences gene regulation and inflammation paves the way for targeted therapeutic strategies.

### 2.7. Genetics, Diversity, and Disparities

Not surprisingly, genetic diversity and health disparities in global human conditions and experiences are related to the differential influences of environmental stresses. The nexus of genetics, diversity, and disparities is intricate in the realm of human health. Genetics, as a fundamental determinant of individual traits, interacts intricately with various factors, including genetic diversity within populations. In immune disorders, this diversity significantly impacts immune responses, affecting susceptibility to autoimmune diseases, infections, and inflammation. Specific genetic variations heighten the risk of autoimmune conditions by influencing immune tolerance mechanisms, emphasizing the need for personalized treatments based on individual genetic profiles [112]. In neurodevelopmental disorders such as ASD, genetics plays a central role, alongside genetic predisposition and environmental factors in ASD development [113]. Population-based genetic diversity leads to various ASD-associated variants, affecting neural development and symptom severity. This interplay underscores the importance of thorough genetic research to uncover mechanisms underlying conditions like ASD, with the potential to enhance diagnostics and targeted therapies [113]. While genetic diversity introduces variations in predisposition to health conditions, it operates within the broader context of diversity and disparities.

In the intricate web of human health encompassing genetics, diversity, and disparities, genetic diversity within populations underscores the multifaceted nature of health outcomes. This diversity extends beyond genetic factors to encompass a rich tapestry of socio-economic, cultural, and environmental variations. These contextual factors wield considerable influence over health disparities, contributing to unequal access to healthcare, resources, and opportunities [114]. The disparities that emerge manifest as varying disease burdens and healthcare outcomes among diverse populations. In the realm of precision medicine, recognizing the role of genetics and its intersection with diversity and disparities is crucial for developing tailored healthcare strategies. Such strategies must address the broader determinants of health while acknowledging the individual genetic nuances within diverse populations, ultimately paving the way for more equitable healthcare and improved health outcomes [115]. 

**Figure 1 ijerph-21-00028-f001:**
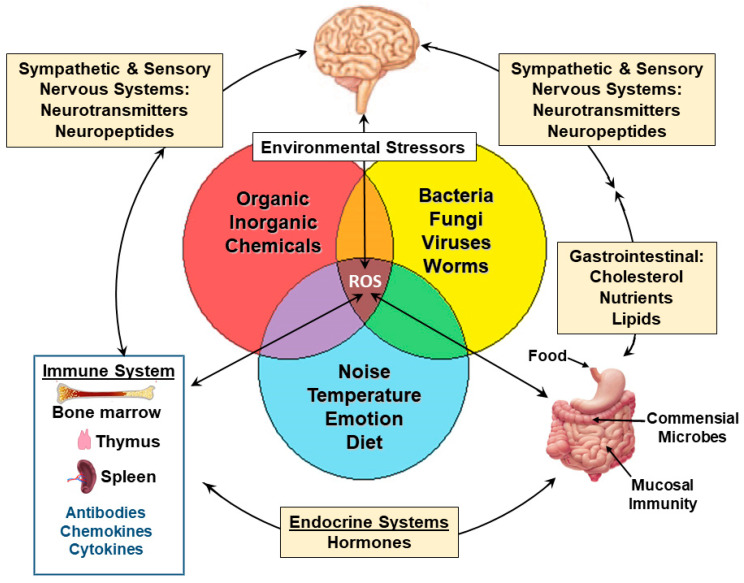
Effects of environmental idiotypes from climate stresses on neuroendocrine, neuroimmune, and gut–brain axis. Multiple forms of environmental stressors induce metabolic syndrome and oxidative stress, and reactive oxygen species (ROS) can create physiological and neurological dysfunctions. Heavy metals such as Pb and Hg as well as norepinephrine (NE) release from the peripheral nervous system cause loss of R-SH in cells, including immune cells, which affect cell function and viability [55,56,57]. The hub of climate stresses and health relates to mitochondrial dysfunction and increased release of ROS when extra levels of ATP are needed. The increased level of ROS modifies the microbiome–gut–brain axis [44,45], the endocrine system, which affects insulin resistance (see Table 1), and immunity.

### 2.8. Mechanistic Connections of Genes and Proteins to Pathologies

A pivotal compilation of insights that bridge the intricate nexus between genetics and a wide spectrum of medical pathologies are shown, connecting genes and mechanisms with certain pathologies (Table 1). Each entry in this comprehensive table offers a profound glimpse into the underlying genetic mechanisms driving the mechanistic connections associated with these diverse diseases. For instance, the inclusion of APOE, APP, PSEN1, and PSEN2 genes in the context of oxidative damage in Alzheimer’s disease presents a compelling illustration of the profound genetic influence on the progression of this debilitating neurodegenerative disorder due to their critical roles in amyloid-beta metabolism, neuronal function, and their impact on the brain’s response to oxidative stress [116,117].

This report presents an overview of genetic associations with various medical conditions, as illustrated (Table 1), shedding light on the intricate genetic underpinnings of these health-related phenomena, from neurodegenerative disorders such as Parkinson’s disease to autoimmune diseases like rheumatoid arthritis, and from metabolic disorders including type 2 diabetes (T2D) to oncology and cardiovascular diseases. Specific health conditions are discussed with some relevant genes and their implications. Parkinson’s disease unveils the mitochondrial dysfunction involvement of *DJ-1*, *PINK1*, and *PARKIN* genes, accentuating the genetic underpinnings of this neurodegenerative condition [118].

Moving into the realm of metabolic disorders, T2D unfolds a complex interplay of *IRS1* and *NFE2L2* genes, influencing insulin resistance and mitochondrial dysfunction [119]. In the case of Type 1 Diabetes (T1D), the dysregulation of Th1/Th2 balance and immune cell activation underscores the genetic basis of autoimmune disease [120]. Oncogenic genetic components TP53, BRCA1, VEGF, and NRF2 preside over DNA damage, angiogenesis, and antioxidant defense mechanisms in various cancers [121]. Cardiovascular disease showcases the oxidative modification of lipids and endothelial dysfunction orchestrated by NOS3, SOD, and HO-1 genes, emphasizing the genetic influences in cardiovascular pathophysiology [122]. With rheumatoid arthritis, there is the underlying role of genetics with TNF-alpha and NF-kB, which orchestrates chronic inflammation [123]. Neurodevelopmental disorders like ASD find their genetic roots in *NLGN3*, *NLGN4*, and *SHANK3* genes, driving neuroinflammation [77]. Systemic Lupus Erythematosus (SLE) features excessive ROS production shaped by genetic factors like IFN-α, IL-1, IL-6, and TNF-α, highlighting the genetic facets of immune dysregulation [124]. Asthma reveals a genetic twist with *GPX1* and *NFE2L2* genes, regulating oxidative stress and inflammation in this respiratory condition [125]. Atherosclerosis points to the oxidative modification of lipids via LDL and scavenger receptors like the macrophage scavenger receptor (SR-A) and CD36 as genetic contributors to vascular pathology [126]. The severity of dysregulation of angiotensin-converting enzyme 2 (ACE2) and the endothelial damage induced by SARS-CoV-2 [127,128] relate to a genetic difference in single-nucleotide polymorphism (SNP) in the 2′-5′ oligoadenylate synthetase (*OAS1*) gene, which modifies early immune defenses against the virus by lowering the recognition of viral RNA [129]. All these genes and products relate directly or indirectly to influences on OS and/or inflammation. Understanding the genetic detriments of OS illuminates intricate disease mechanisms and paves the way for personalized medicine by pinpointing relevant genetic markers and pathways.

### 2.9. Metabolic Syndrome, Lipid Dysregulation, and Immune Activity

Roberts and Sindhu [130] describe the intricate relationship between metabolic syndrome, lipids, and immune activity. The hallmark of dyslipidemia associated with metabolic syndrome is characterized by elevated triglycerides and diminished HDL cholesterol levels, which emphasizes the lipid-related components of this syndrome. OS, a pivotal aspect of metabolic syndrome, exerts a profound influence on immune activity. This association highlights the intricate interplay between metabolic dysfunction, lipid metabolism, and immune responses, underscoring the complexity of this multifaceted syndrome and its broader implications for human health.

With disturbances in lipid metabolism, metabolic syndrome and OS contribute to the development of conditions like atherosclerosis, hypertension, and T2D disorders; however, OS is both an early event in their pathogenesis as well as a consequence of these health problems [130,131]. Table 2 and Table 3 describe the associations of metabolic syndrome and dyslipidemia with the induction of OS and how, together, they affect certain pathologies. Notably, individuals with metabolic syndrome exhibit indications of increased oxidative damage, including diminished antioxidant defenses, reduced levels of vitamins such as vitamin C and α-tocopherol, decreased activity of antioxidant enzymes like superoxide dismutase, and elevated levels of markers such as malondialdehyde, protein carbonyls, and xanthine oxidase activity [130].

Some studies have explored the potential benefits of supplements like γ- and α-tocopherol or drugs like allopurinol, which inhibits xanthine oxidase, in reducing OS levels occurring with metabolic syndrome [130]. Fat accumulation, especially around the waist, appears to be positively associated with OS-related problems, including endothelial dysfunction [130,132,133]. Metabolic disorders are associated with innate and adaptive immune cells in adipose tissue creating systemic inflammation and insulin resistance [134]. The associated involvement of intestinal microbes adds fuel to the fire [135,136]. Overall, OS appears to be intimately intertwined with the various components of metabolic syndrome, clarifying the complex mechanisms underlying this multifaceted condition. Mitochondrial dysfunction is also intimately connected with metabolic syndrome and OS [137,138]. 

The lipid-related aspect of metabolic syndrome, dyslipidemia with elevated triglycerides, is crucial in understanding its pathophysiology. One key mechanistic insight lies in the relationship between dyslipidemia and immune dysfunction. Elevated triglycerides can lead to free fatty acid accumulation and lipid metabolites in various tissues, including the liver and adipose tissue. This accumulation triggers a series of events, including increased OS, inflammation, and the attraction and activation of immune cells like macrophages. The activation of these immune cells in response to lipid overload is mediated by various signaling pathways, such as toll-like receptor (TLR) signaling, and leads to the production of pro-inflammatory cytokines. OS is a critical component of metabolic syndrome, and it further exacerbates immune dysfunction. As shown in Table 2, OS and metabolic syndrome often coexist due to chronic inflammation and can be exacerbated by factors like dyslipidemia. OS can damage cell membranes and DNA, leading to the release of damage-associated molecular patterns (DAMPs) that activate immune responses. Additionally, OS can modulate signaling pathways involved in immune cell activation and cytokine production.

The mechanistic understanding of the complex interplay between dyslipidemia, OS, and immune activity in the context of metabolic syndrome highlights the mechanistic pathway. It elucidates how dyslipidemia-induced inflammation, coupled with OS-induced immune activation, contributes to the progression of metabolic syndrome, as summarized in Table 2. These insights provide a foundation for developing targeted therapeutic approaches to mitigate the adverse effects of metabolic syndrome on human health.

OS serves as a common link connecting metabolic syndrome with lipid dysregulation and immune activity [133]. OS, a central feature of metabolic syndrome, can worsen dyslipidemia by promoting the formation of ROS and lipid peroxidation. These metabolic changes cause cell stress, which induces ER stress, the unfolded protein response (UPR) [138], and the activation of Hsp90 chaperones [139]. The Hsp90 chaperones include the cytosolic Hsp90, Grp94 of the ER, and mitochondrial Trap-1 [140]. These HSP90 proteins modulate organelle machinery including transcription; Trap-1 activity connects again with mitochondrial dysfunction [141], and in the brain, mitochondrial chaperones affect neuropathologies [10,142]. Many different co-chaperones bring substrate (“client” proteins) to Hsp90 to affect the folding, function, and stability of the clients. A list of the co-chaperones and the clients is maintained online (http://www.picard.ch/downloads/downloads.htm (accessed on 30 October 2023)). Although there are physiological attempts to control stress, e.g., the expression of chaperones and co-chaperones, the system can be overwhelmed by DAMPs, ROS, and lipid radicals skewing adaptive immunity away from pathogens and toward self- and altered-self constituents. Furthermore, chronic inflammation alone from innate immune cells, which is often observed with metabolic syndrome, can disrupt developmental and regulatory pathways.

**Table 2 ijerph-21-00028-t002:** Interconnections between metabolic syndrome, dyslipidemia, and OS.

Factor	Connection with Metabolic Syndrome	Connection with Dyslipidemia	Connection with OS	Reference
Environmental stressors	Exacerbates Metabolic Syndrome by increasing stress hormones and inflammation.	May lead to dyslipidemia through dietary and lifestyle factors.	Can trigger oxidative stress through the release of free radicals in response to stress.	Masenga et al. [143] Fahed et al. [144]
Metabolic Syndrome	-	Dyslipidemia is a common component of Metabolic Syndrome.	Oxidative stress often accompanies Metabolic Syndrome due to chronic inflammation.	Huang [145]
Dyslipidemia	Dyslipidemia is a hallmark feature of Metabolic Syndrome.	-	Dyslipidemia can contribute to oxidative stress through the production of reactive oxygen species (ROS).	Tangvarasittichai [131]
Oxidative Stress	Oxidative stress is closely linked to Metabolic Syndrome, contributing to its progression.	Can be exacerbated by dyslipidemia due to increased ROS production.	-	Roberts and Sindhu [130]

**Table 3 ijerph-21-00028-t003:** Pathological associations with metabolic syndrome.

Pathology	Association with Metabolic Syndrome	Reference
Dyslipidemia	Dyslipidemia is a common component of metabolic syndrome, characterized by elevated triglycerides and low HDL cholesterol.	Pappan and Rehman [146]
Oxidative Stress	Metabolic syndrome is often accompanied by oxidative stress, marked by increased levels of reactive oxygen species (ROS).	Monserrat-Mesquida et al. [147]
Inflammation	Chronic inflammation is a hallmark of Metabolic Syndrome and contributes to its progression.	Pahwa et al. [148]
Insulin Resistance	Metabolic Syndrome is closely linked to insulin resistance, which leads to elevated blood sugar levels.	Fahed et al. [144]
Hypertension	Hypertension often co-occurs with metabolic syndrome and contributes to cardiovascular risks.	Stanciu et al. [149]

**Figure 2 ijerph-21-00028-f002:**
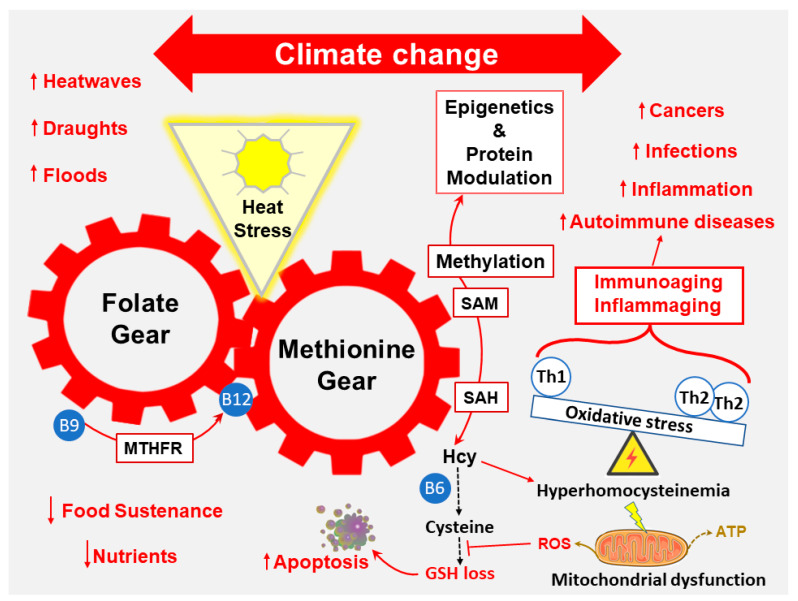
Metabolic disruption and oxidative stress associated with climate changes. Changes to climate have brought more and longer impacts of heat stress which disrupts folate and methionine cycling leading to oxidative stress (OS), mitochondrial dysfunction (reactive oxygen species (ROS) > ATP), dyslipidemia, and hyperhomocysteinemia. The levels of nutrients such as B vitamins and antioxidants are needed to prevent these disruptions from metabolic syndrome, elevated cell and molecular damage, and alteration of immune activities (Th2 > Th1, Th17 > Treg, Neutrophils > lymphocytes). Heat stress along with other climate-related stressors can lead to a multitude of different pathologies. Older individuals and those with certain co-morbidities are more susceptible to the stressors. OS occurs when ROS > glutathione (GSH). The upward and downward arrows represent increased or decreased, respectively.

**Figure 3 ijerph-21-00028-f003:**
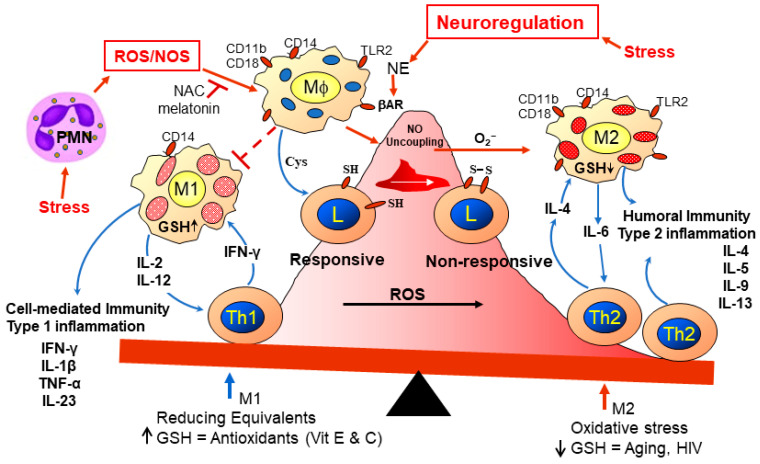
Redox imbalance diminishes Th1 host defenses and may promote allergies, asthma, and autoimmune diseases with type 2 inflammation. When antigen-presenting cells like dendritic cells and macrophages have lower GSH levels (M1 cells), they cannot enhance Th1 activation to kill viruses as occurs with HIV, but the M2 subpopulation can still aid activation of Th2 until the level of GSH diminishes to the level of that occurring with AIDS. Melatonin and N-acetylcysteine (NAC) can inhibit ROS and nitrogen oxygen species (NOS), which, in part, are released by polymorphonuclear cells (PMN). Increased (↑) GSH levels provide more antioxidant activity and aid M1-Th1 interactions for cell-mediated immunity including cytolytic CD8 T cell functions; whereas, the downward arrow indicates loss of GSH which initially allows M2-Th2 interactions.

### 2.10. Environmental Stressors’ Influence on Metabolic Syndrome: A Complex Interplay

Expanding on the factors previously discussed, Table 4 provides insights into specific pathologies associated with metabolic syndrome with attention paid to environmental stressors, which can significantly influence the development and exacerbation of metabolic syndrome through various mechanisms. Climate change and temperature rise, exacerbate metabolic dysfunction by subjecting individuals to heat stress, disrupting sleep patterns, and fostering increased inflammation. Psychological stress, another noteworthy factor, can induce hormonal changes and unhealthy coping behaviors, ultimately contributing to the onset of metabolic syndrome. Additionally, diet and nutrition play a pivotal role, as poor dietary choices in response to stressors can further compound the risk of metabolic syndrome. These stressors collectively underscore the intricate interplay between environmental factors and the complex pathogenesis of metabolic syndrome, emphasizing the importance of addressing both physical and psychological well-being to mitigate its impact.

## 3. Conclusions

Researchers are actively engaged in exploring the complex interplay of environmental stressors, genetics, and dietary influences on human health. Many studies have explored the connections between environment and diverse health conditions, illuminating the intricate interactions among climate change, chemical exposures, and susceptibility to different diseases. Additionally, research endeavors have delved into the role of OS as a potential mechanistic link between environmental factors and neural and immune-related disorders, paving the way for a deeper future understanding of the pathophysiological processes underlying these disorders. Studies examining the influence of nutritional elements, notably B vitamins, genetic variations within the MTHFR gene, and the methionine cycle enzymes on hyperhomocysteinemia and immune dysfunction have yielded valuable insights into potential indicators for diagnosis and targets for treatment. Related studies collectively contribute to a growing body of knowledge that not only underscores the significance of the environment in shaping health outcomes but also offers promising avenues for improving diagnosis and treatment strategies for neural conditions like Alzheimer’s disease and ASD and immunopathological mechanisms induced by pathogens such as SARS-CoV-2.

The complex interplay among environmental factors, genetics, and nutrition with a primary emphasis on the significant implications of climate change, particularly its association with oxidative stress and heatwaves, highlights the undeniable connection between the environment and human well-being. The environment, with its biological, chemical, physical, and psychological stresses affecting the common detriment of OS-altered immunity, shapes our health status throughout the lifespan. Climate change introduces new challenges to our health landscape. It not only alters disease patterns and increases disease transmission risks but also reshapes ecosystems, affecting plant growth patterns and exacerbating food supply challenges and respiratory allergies. If climate changes are allowed to continue, these anxieties will increase, gradually eroding mental well-being in combination with physical health.

## 4. Future Directions

A central point focuses on OS and its critical impact on health. Repeated exposure to heat stress, exacerbated by climate change, demands higher energy expenditure, resulting in elevated levels of ROS and subsequently leading to OS. To confront the environmental stresses, improved means to control mitochondrial dysfunction and metabolic syndrome are needed. Improved control of the interconnected network of the gut-brain axis and neuroendocrine immune pathways will lessen leaky gut syndrome and the levels of OS and inflammation. Genetics and the environment combine to influence health patterns. To understand the rising prevalence of health conditions like allergies, autoimmune disorders, neuropathologies, and the type 2 inflammation associated with these variant disorders/diseases over the past few decades, we must better understand system biology and the multitude of molecular intra- and inter-cellular controls such as chaperones, co-chaperones, and their clients, as well as cellular variances dependent on their neighborhood. The importance of genetics, diversity, and disparities in understanding the complexities of health issues also highlights the need for personalized and equitable healthcare solutions. The recent MIS-C and Long-Covid disorders also offer potential directions for diagnosis and treatment strategies for diseases affecting systems biology. The different forms of environmental stress have a common condition (OS-altered immunity) that shapes our health status from fetuses to elderly adults. A central theme explored is the multi-faceted influence of climate change, driven by chemical, biological, physical, and psychological stresses. Climate change introduces new challenges to our health landscape; it can alter disease patterns and increase disease transmission risks but also reshapes ecosystems, affecting plant growth patterns and exacerbating food supply challenges and respiratory allergies. A vivid picture is painted describing stressors gradually eroding mental well-being in combination with physical health.

A central point of our discussion focuses on OS and its critical impact on health. Repeated exposure to heat stress, exacerbated by climate change, demands higher energy expenditure, resulting in elevated levels of ROS. This intricate process compromises cellular functions and can potentially result in cell death. The interconnected network of the gut–brain axis and neuroendocrine immune interactions is also impacted, potentially contributing to leaky gut syndrome and enhanced inflammation.

The article further dives into the intricate folate and methionine cycles, revealing how they are linked to autoimmune disorders, inflammation, and complex health conditions such as ASD and MIS-C. Elevated homocysteine levels, termed hyperhomocysteinemia, are implicated in both conditions, presenting a promising avenue for diagnostics and therapeutic interventions. The review also touches upon the rising prevalence of health conditions like allergies, autoimmune disorders, and type 2 inflammation, and their increased prevalence. The importance of genetics, diversity, and disparities in understanding these complex health issues highlights the need for personalized and equitable healthcare solutions.

## Figures and Tables

**Table 1 ijerph-21-00028-t001:** Mechanistic connections of genes to OS and pathologies.

Pathology	Mechanistic Connection	Gene/Protein	Reference
Alzheimer’s Disease	Oxidative damage	APOE, APP, PSEN1, PSEN2	Buccellato et al. [116] Luca et al. [117]
Parkinson’s Disease	Mitochondrial dysfunction	DJ-1, PINK1, PARKIN	Moon and Paek [118]
Type 2 Diabetes	Insulin resistance, Mitochondrial Dysfunction	IRS1, Nrf2	Shehata [119]
Type 1 Diabetes	Dysregulation of Th1/Th2 balance and immune cell activation; autoimmunity	Th1 and Th2 cells Tumor Necrosis Factor-alpha (TNF-α)	Vaseghi and Jadali [120]
Cancer	DNA damage, Angiogenesis Antioxidant Defense	TP53, BRCA1, VEGF, NRF2	Moon et al. [121]
Cardiovascular Disease	Oxidative modification of lipids and endothelial dysfunction	NOS3, SOD, HO-1	Sun et al. [122]
Rheumatoid Arthritis	Inflammation	TNF-alpha, NF-kB, etc.	Farrugia and Baron [123]
Autism Spectrum Disorder (ASD)	Neuroinflammation	NLGN3, NLGN4, SHANK3, etc.	Pangrazzi et al. [77]
Systemic Lupus Erythematosus (SLE)	Excessive ROS production	IFN-α, IL-1, IL-6, and TNF-α	Ameer et al. [124]
Asthma	Regulation of oxidative stress and inflammation	GPX1 (Glutathione Peroxidase 1), NFE2L2 (Nuclear Factor Erythroid 2 Like 2)	Fitzpatrick et al. [125]
Atherosclerosis	Oxidative Modification of Lipids	Low-density Lipoproteins (LDL) Scavenger Receptors (e.g., SR-A, CD36)	McMahon and Hahn [126]
Severe Acute Respiratory Syndrome (SARS)	Endothelial Dysfunction Coagulation Imbalance ACE2 Receptor Modulation	Endothelin-1 (ET-1) Plasminogen Activator Inhibitor-1 (PAI-1) Angiotensin-Converting Enzyme 2 (ACE2) Angiotensin II (Ang II)	Mehrabadi et al. [127] Vassiliou et al. [128]

**Table 4 ijerph-21-00028-t004:** Pathological associations with metabolic syndrome references.

Environmental Stressor	Effect on Metabolic Syndrome	Reference
Climate Change and Temperature Rise	May exacerbate metabolic dysfunction through heat stress, disrupted sleep, and increased inflammation.	Natour et al. [150]
Psychological Stress	Can lead to hormonal changes and unhealthy coping behaviors, contributing to metabolic syndrome.	Ryan. [151]
Diet and Nutrition	Poor dietary choices in response to stressors can contribute to metabolic syndrome.	Castro-Barquero et al. [152]

## Data Availability

Data are contained within the article.

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
