# Peer review of "Climate Stressors and Physiological Dysregulations: Mechanistic Connections to Pathologies"

_ijerph, 2023, doi:10.3390/ijerph21010028_

Round 1
Reviewer 1 Report
Comments and Suggestions for Authors
Reviewer report: The Clinical and Environmental Nexus: Pathological Associations of Stressors, Metabolic Syndrome, Genetics, and Immune 3 Dysregulation with Climate Changes
Overall
This is a wonderful piece of work, and I can see it becoming an important reference article for people trying to understand this complex web of environment and health. Thank you for the opportunity to review. I have no major comments, and just some minor items that could be addressed.
Minor comments
Line 71: ASD is not a neurodegenerative disease, and should be separated from this statement.
Line 155: Please explain what is meant by cold restraint.
Line 162: This sentence is a little awkward. Perhaps: ‘…consider environmental stressors in the development of allergies….’.
Line 162-168: References required for this section.
Line 169-172: I find the comparison of the Th1/2 balance to HIV/AIDS a little jarring. What is the relevance and are they really a direct comparison?
Line 269: It is not immediately clear why the MTHFR gene has been chosen as an example. What is the environmental link with this gene?
Author Response
Reviewer 1
Line 71: ASD is not a neurodegenerative disease, and should be separated from this statement. This has been corrected by changing ASD to a neurodevelopmental disorder.
Line 155: Please explain what is meant by cold restraint. “Cold restraint” method is now described.
Line 162: This sentence is a little awkward. Perhaps: ‘…consider environmental stressors in the development of allergies….’. Allergies are included with the other pathologies; this is addressed more in response to the next two concerns.
Line 162-168: References required for this section. See response to next concern
Line 169-172: I find the comparison of the Th1/2 balance to HIV/AIDS a little jarring. What is the relevance and are they really a direct comparison? The section on type 2 inflammation and Th1 and Th2 cells has been slightly expanded. Environmental stresses affecting allergies is included. To further help in explaining the Th1 and Th2 differences we also have added a new figure (Figure 3) to show oxidative stress on Th1: Th2 balance with inclusion of cytokine from M1 and Th1 versus those from M2 and Th2. We hope this addresses all concerns related to types of immunity with Th1 and Th2. A new reference for type 2 inflammation also was added. Regarding HIV and shift toward Th2 this has been well documented for many years and it is references [65,66]. We now how also provided a better reference for type 2 inflammation
Line 269: It is not immediately clear why the MTHFR gene has been chosen as an example. What is the environmental link with this gene? We have added a sentence that connects with Figure 2. Individuals with mutated forms of MTHFR are known to be more susceptible to some conditions. We are suggesting hear stress from climate is an inducer of problems to the folate cycle which is needed for the methionine cycle which helps to drive methylation of DNA and proteins.
Reviewer 2 Report
Comments and Suggestions for Authors
This submission is a review that focuses on the adverse health effects of oxidative stress due to climate change, an interesting topic on increasing importance to the health sciences. That said, there are major changes that are needed prior to publication.
Overall, a more pointed review in relation to the adverse health effects due to climate change is needed. At times (e.g. section 2.3) the manuscript becomes a typical review of oxidative stress – induced adverse health or cellular effect X, Y and Z, which has been previously published. An effort to keep the focus of the review on cellular effects directly related to climate change would improve the submission.
Lines 104 – 132 is redundant summary of the previous Introduction section. This needs to be more concise and avoid repeating what was stated in the Introduction.
Line 160; define type II inflammation
Line 164; define type II immunity
Line 166 ASD needs to redefined
Line 178 Neutrophil-Lymphocyte Ratio is given the acronym NLR but does not appear later in the review. Throughout the manuscript, many acronyms were defined initially but not used subsequently. If a protein, gene or ratio, etc… is referenced only once then providing an acronym for it is unnecessary.
Author Response
At times (e.g. section 2.3) the manuscript becomes a typical review of oxidative stress – induced adverse health or cellular effect X, Y and Z, which has been previously published. An effort to keep the focus of the review on cellular effects directly related to climate change would improve the submission. Oxidative stress is a major aspect of our review and it has been emphasized regarding the folate/methionine cycles as emphasized in Figures. It is the oxidative stress which lowers the glutathione level in cells and in plasma. Loss of glutathione has a profound influence on the Th1 and Th2 levels. Lees Th1 will increase susceptibility to infections and cancers. More Th2 can lead to more antibody production including autoantibodies. More autoantibodies can cause more inflammation (type 2 inflammation) with can activate more eosinophils to enhance skin rashes, allergies and asthma.
Lines 104 – 132 is redundant summary of the previous Introduction section. This needs to be more concise and avoid repeating what was stated in the Introduction. Some of the issues discussed are similar. Sometimes points need to be restated in different ways to emphasize the concerns being made. We have made some modification lessen overlap.
Line 160; define type II inflammation We have expanded on the type2 immunity and inflammation and added a reference. We also have tried to make this more clear with inclusion of Figure 3.
Line 164; define type II immunity Addressed with additional information on type 2 inflammation. Many autoimmune disease as well as allergies and asthma are due to type 2 immunity.
Line 166 ASD needs to redefined As requested by Reviewer 1 we have clarified the ASD
Line 178 Neutrophil-Lymphocyte Ratio is given the acronym NLR but does not appear later in the review. Throughout the manuscript, many acronyms were defined initially but not used subsequently. If a protein, gene or ratio, etc… is referenced only once then providing an acronym for it is unnecessary. This has been corrected
Reviewer 3 Report
Comments and Suggestions for Authors
General comment:
The review manuscript entitled “The Clinical and Environmental Nexus: Pathological Associations of Stressors, Metabolic Syndrome, Genetics, and Immune Dysregulation with Climate Changes” brings useful information regarding the complex relationship between climate change and human health. The authors tried to explore the complex relationship of environmental factors, stressors, their mechanistic cellular and molecular effects, and their significant impact on human health due to climate change. The reviewer believed that the present study is interesting and potentially could contribute to the research field as it tried to address very crucial issues of climate change and its effects on human health, disease pathology, and molecular mechanisms. However, some concerns must be addressed to clarity and improve the present version of the manuscript. It can be published after taking care of some issues as follows;
1. Title: The title seems informative and relevant to the major findings of the review, but a bit long. Therefore, title shortening/rewriting is recommended.
2. Abstract: It is written in a diluted way. It should have a hypothesis and clearly mentioned objectives along with a short methodological touch.
3. Introduction: The research gap/question is not clearly outlined. The authors should emphasize discussing the novelty and significance of the study. There are already several review articles available on similar topics, so why this review article is important? What is the difference and new finding from other review papers? Please try to explain carefully.
4. A methodology section must be included where the search strategy, exclusion and inclusion of articles and their criteria must be described. A good review should clearly indicate the scope of the review, the criteria utilized to determine whether or not papers were to be included or excluded from the review, and a summary of previous review papers (thereby indicating whether or not, or to what extent, there is a gap and where or if a further review is required).
However, the review manuscript is well written and just needs to address the above-mentioned concerns.
Author Response
Title: The title seems informative and relevant to the major findings of the review, but a bit long. Therefore, title shortening/rewriting is recommended.
Response: As recommended the title length has been shortened, we have shortened the title and hope that it is now consider acceptable.
Abstract: It is written in a diluted way. It should have a hypothesis and clearly mentioned objectives along with a short methodological touch. The abstract has been modified and it includes our hypothesis.
Response: The abstract has been modified to include hypothesis and objectives. Methods have been added to the Introduction.
Introduction: The research gap/question is not clearly outlined. The authors should emphasize discussing the novelty and significance of the study. There are already several review articles available on similar topics, so why this review article is important? What is the difference and new finding from other review papers? Please try to explain carefully.
Response: The introduction now indicates aspects of novelty and significance.
A methodology section must be included where the search strategy, exclusion and inclusion of articles and their criteria must be described. A good review should clearly indicate the scope of the review, the criteria utilized to determine whether or not papers were to be included or excluded from the review, and a summary of previous review papers (thereby indicating whether or not, or to what extent, there is a gap and where or if a further review is required).
Response: We have added a section about literature searching method. At end of Introduction.
Round 2
Reviewer 2 Report
Comments and Suggestions for Authors
Authors did a good job in addressing all the reviewers’ comments.